# Effect of 1α,25(OH)_2_ Vitamin D_3_ in Mutant P53 Glioblastoma Cells: Involvement of Neutral Sphingomyelinase1

**DOI:** 10.3390/cancers12113163

**Published:** 2020-10-28

**Authors:** Samuela Cataldi, Cataldo Arcuri, Andrea Lazzarini, Irina Nakashidze, Francesco Ragonese, Bernard Fioretti, Ivana Ferri, Carmela Conte, Michela Codini, Tommaso Beccari, Francesco Curcio, Elisabetta Albi

**Affiliations:** 1Department of Pharmaceutical Sciences, University of Perugia, 06126 Perugia, Italy; samuelacataldi@libero.it (S.C.); irinanakashidze@yahoo.com (I.N.); carmela.conte@unipg.it (C.C.); michela.codini@unipg.it (M.C.); tommaso.beccari@unipg.it (T.B.); 2Department of Experimental Medicine, University of Perugia, 06126 Perugia, Italy; cataldo.arcuri@unipg.it; 3CRABiON, 06100 Perugia, Italy; andrylazza@gmail.com; 4Department of Chemistry, Biology and Biotechnologies, Via Elce di Sotto 8, University of Perugia, 06126 Perugia, Italy; francescoragonese85@gmail.com (F.R.); bernard.fioretti@unipg.it (B.F.); 5Division of Pathological Anatomy and Histology, University of Perugia, 06126 Perugia, Italy; ivanaferri@gmail.com; 6Department of Medicine (DAME), University of Udine, 33100 Udine, Italy; francesco.curcio@uniud.it

**Keywords:** glioblastoma, neutral sphingomyelinase, vitamin D3, vitamin D receptor, U251

## Abstract

**Simple Summary:**

The active form of vitamin D3 slows proliferation and stimulates differentiation of glioblastoma cells with mutated p53 by acting via neutral sphingomyelinase 1 and not by classical via vitamin D receptor-mediated

**Abstract:**

Glioblastoma is one the most aggressive primary brain tumors in adults, and, despite the fact that radiation and chemotherapy after surgical approaches have been the treatments increasing the survival rates, the prognosis of patients remains poor. Today, the attention is focused on highlighting complementary treatments that can be helpful in improving the classic therapeutic approaches. It is known that 1α,25(OH)_2_ vitamin D_3_, a molecule involved in bone metabolism, has many serendipidy effects in cells. It targets normal and cancer cells via genomic pathway by vitamin D3 receptor or via non-genomic pathways. To interrogate possible functions of 1α,25(OH)_2_ vitamin D3 in multiforme glioblastoma, we used three cell lines, wild-type p53 GL15 and mutant p53 U251 and LN18 cells. We demonstrated that 1α,25(OH)_2_ vitamin D3 acts via vitamin D receptor in GL15 cells and via neutral sphingomyelinase1, with an enrichment of ceramide pool, in U251 and LN18 cells. Changes in sphingomyelin/ceramide content were considered to be possibly responsible for the differentiating and antiproliferative effect of 1α,25(OH)_2_ vitamin D in U251 and LN18 cells, as shown, respectively, in vitro by immunofluorescence and in vivo by experiments of xenotransplantation in eggs. This is the first time 1α,25(OH)_2_ vitamin D3 is interrogated for the response of multiforme glioblastoma cells in dependence on the p53 mutation, and the results define neutral sphingomyelinase1 as a signaling effector.

## 1. Introduction

Glioblastoma multiforme (GBM) is the most common and one of the most aggressive primary intracerebral tumors with great heterogeneity in tissue histology and imaging [1]. The standard initial approach is maximal safe surgical resection, followed by radio- and chemotherapy, and treatment options in the recurrent setting are less well defined, with no established standard of care [2]. Some molecules have recently been identified that can influence the development of glioblastoma and the course of the disease, such as obtusaquinone [3], coronarin D [4], betulinic acid [5], icaritin [6], and others. As early as 1995 Magrassi et al. [7] had demonstrated that activated 1α,25(OH)_2_ vitamin D_3_ or calcitriol (1α,25(OH)_2_VD3) was capable of inducing a significant (>50%) reduction in growth of glioblastoma cells at dosages over 5 μM. Classically, the inactive form of vitamin D3 or cholecalciferol is first hydroxylated in carbon 25 in the liver to form 25(OH) vitamin D_3_ or calcidiol and then it is hydroxylated in carbon 1 in the kidney to form 1α,25(OH)_2_VD3. Interestingly, glioblastoma was able to metabolize cholecalciferol to calcidiol [8]. It has been demonstrated that five cell lines of GBM, asTx3095, Tx3868, U87, U118, U373, were resistant against the antiproliferative activity of 1α,25(OH)_2_VD3 at the dosage of 10^−6^, 10^−8^, and 10^−10^ M [9]. Differently, an analog of 1α,25(OH)_2_VD3, diethyl [(5*Z*,7*E*)-(1*S*,3*R*)-1,3-dihydroxy-9,10-secochola-5,7,10(19)-trien-23-in-24-yl] phosphonate analogue (EM1), exerted in U251 and T98G GBM cell lines anti-migratory effects and decreased their invasive capacity [10]. Moreover, other 1α,25(OH)_2_VD3 analogs, such as tacalcitol and calcipotriol, inhibited the proliferation and migration of T98G [11]. McConnel et al. demonstrated that 1α,25(OH)_2_VD_3_ had a slight temozolomide synergism for U87-MG [12]. This synergism was supported by Elmaci et al. who showed that the addiction of 1α,25(OH)_2_VD_3_ totemozolomide reduced mortality in GBM patients in comparison to nonusers [13]. Thus, despite several studies suggesting that 1α,25(OH)_2_VD_3_ is essential for cell signaling in the nervous system and is involved in the suppression of several cancer cell proliferation and migration [14], the potential mechanisms for these responses remain unclear.

It is known that the effects of 1α,25(OH)_2_VD_3_ on target cells are mediated via binding to the nuclear vitamin D receptor (VDR), a ligand-dependent transcription factor [15] located in inner nuclear membrane microdomains [16]. VDR is expressed in many normal tissues, including in adult and embryonic cells of nervous system [17,18]. However, increasing evidence provides information on the involvement of nongenomic pathways in the response to 1α,25(OH)_2_VD3 [8,19]. Interestingly, VD3 and/or 1α,25(OH)_2_VD3 were able to induce apoptosis or cell growth delay, with subsequent cell differentiation, thanks to the degradation of sphingomyelin (SM) with the production of ceramide, a molecule involved in cell signaling [20,21]. 

SM is a structural and bioactive lipid that is hydrolyzed to ceramide by sphingomyelinases (SMase) classified based on their activity pH optima into alkaline, acid, and neutral subtypes [22]. Alkaline SMase (alkSMase) is present at intestinal level and is useful for dietary SM; acid SMase (aSMase) and neutral SMase (nSMase) are present in almost all cells and play fundamental roles in several pathophysiological conditions [23,24]. aSMase isoform is located in lysosome and it is encoded by the *SMPD1* gene. The nSMase family includes four isoforms, nSMase1, nSMase2, nSMase3, and nSMase4 encoded by four different genes *SMPD2*, *SMPD3*, *SMPD4*, *and SMPD5*, respectively [25]. nSMase1 is located in the reticulum endoplasmic/Golgi apparatus and in the cell nucleus and it was involved in apoptosis and cancer; nSMase2 is specific of the plasma membrane and it was involved in exosome formation, and in the inflammatory response; nSMase3 is located in endoplasmic reticulum and it was involved in cellular stress response; nSMase4 belongs to the mitochondria and its function is not yet clear in human cells [25]. In cancer: (a) aSMase was able to reprogram the tumor immune microenvironment [26] and to participate in apoptotic cell death [27,28]; (b) nSMase1 had both pro- and anti-cancer roles [29].

In specific cancer cells that did not express VDR, such as gastric cancer cells, 1α,25(OH)_2_VD3 induced apoptosis by upregulating the gene and protein expression of aSMase [28]. Differently, when VD3, and consequently its metabolite 1α,25(OH)_2_VD3 were responsible for cancer cell differentiation they required nSMase enzyme, as occurred for leukemic HL-60 cells thanks to the production of ceramide as lipid mediator [30].

The decrease in SM and the increase in ceramide content upon 1α,25(OH)_2_VD3 treatment in glioblastoma cells was shown about twenty years ago [31] but later the research was not followed. The study was only an observation of lipid variation and with really high 1α,25(OH)_2_VD3 concentrations (5 μM), which could be toxic to normal cells present in vivo. Furthermore, in those days the SMase isoforms were not known and lipidomics studies did not exist. However, the research was promising considering the different responses to 1α,25(OH)_2_VD3 treatment in various cell types of GBM above reported [8,9,10,11,12,13,14]. 

Therefore, in the present study, we aimed to investigate the mechanism of action of 1α,25(OH)_2_VD3 in three cell types of glioblastoma cells: wild-type p53 GL15 cells, and mutant p53 U251 and LN18 cells. Since p53 protein is a tumor suppressor protein inducing tumor cell apoptosis [32], U251 and LN18 cells are less exposed to apoptosis than GL15 cells. Thus, we studied a genomic response via VDR and a non-genomic response via SMase in the two cell models. Because of their importance in cancer [26,27,28,29], we focused the study on aSMase and nSMase1. This work might support the use of 1α,25(OH)_2_VD3 in GBM reported in literature as a valid implementation to traditional therapy.

## 2. Results

In order to exclude the possible cytotoxic effect of 100 and 400 nM 1α,25(OH)_2_VD3 in GL15, U251, and LN18 cells, we used MTT assay after 24 h in cell culture. The results show no significant differences in the viability of GL15, U251, and LN18 cells cultured in the presence of 100 and 400 nM with respect to control cells, by excluding the cytotoxic effect of 1α,25(OH)_2_VD3 at the doses used for the subsequent experiments. 1% and 2% DMSO were used as positive controls (Figure 1).

Since 1α,25(OH)_2_VD3 stimulates the gene expression of himself receptor [33], to investigate the role for 1α,25(OH)_2_VD3 in GBM, we began by studying whether GL15, U251, and LN18 cells expressed VDR gene and protein. We analyzed VDR after treatment with a physiological dose of vitamin (minimum limit, 100 nM), and with a high dose (about 10% above maximum limit, 400 nM) [18], considering the possibility that cancer cells could be resistant to 1α,25(OH)_2_VD3 treatment. As expected, in GL15 100 nM 1α,25(OH)_2_VD3 was able to upregulate VDR (Figure 2). Surprisingly, by increasing vitamin concentration, VDR did not change (Figure 2). To translate this finding into an experimental model of mutated p53 tumor suppressor gene, we performed the same study in U251 and LN18 cells. The data showed a trend of not change of VDR with 100 nM 1α,25(OH)_2_VD3 and a slight downregulation with 100 nM 1α,25(OH)_2_VD3 (Figure 2). 

Based on the data indicating different responses of VDR to 1α,25(OH)_2_VD3 treatment in the cell lines with mutant p53 (U251 and LN18) respect to GL15 cells, we elected to analyze aSMase and nSMase1 enzymes to define the relationship between 1α,25(OH)_2_VD3/VDR and SM breakdown. We validated the effect of the vitamin by analyzing the expression of genes encoding for the two enzymes, *SMPD2* for nSMase1 and *SMPD1* for aSMase. In GL15 cells, 1α,25(OH)_2_VD3 induced a low *SMPD2* overexpression only at the concentration of 400 nM and a low down expression of *SMPD1* only at the concentration of 100 nM (Figure 2). In U251 cells, a strong increase in *SMPD2* expression 1α,25(OH)_2_VD3 concentration-dependent and a low decrease in *SMPD1* expression with 100 nM were observed (Figure 2). The same response to 1α,25(OH)_2_VD3 treatment was evident in LN18 cells, albeit with slightly lower values (Figure 2).

Likewise, we assessed VDR, nSMase1 and aSMase protein levels through immunoblot analysis (Figure 3 and Appendix A). The results indicate that VDR protein was predominantly upregulated in GL15 cells treated with 100 nM 1α,25(OH)_2_VD3 and nSMase1 was predominantly upregulated in U251 and LN18 cells treated with 400 nM 1α,25(OH)_2_VD3, according to the data of gene expression. These results led us to hypothesize that 1α,25(OH)_2_VD3 at high doses used a non-genomic pathway involving the nSMase1 in U251 cells. 

Thus, we sought to determine whether changes of nSMase1 protein level in U251 cells caused by 400 nM 1α,25(OH)_2_D_3_ were consistent with variation of enzyme activity. As you can see, the nSMase activity was significantly higher in U251 treated with 1α,25(OH)_2_D_3_ than control samples (Figure 4a panel A). The same effect was evident in LN18 cells (Figure 4a panel B). To corroborate the data, we performed a lipidomic analysis in untreated and 1α,25(OH)_2_D_3_ treated U251 and LN18 cells by using 12:0 SM, 16:0 SM, 18:1 SM,24:0 SM, Cer 16:0, Cer 20:0, Cer 24:0 external calibrators, in order to further understand the effect of the enzyme activation in SM breakdown. The results highlight that 1α,25(OH)_2_D_3_ significantly reduced 16:0 SM and 24:0 SM species (Figure 4b panel A and B) and increased 16:0 ceramide and 24:0 ceramide species (Figure 4c panel A and B). To have a deeper insight into SM species in order to know the presence of saturated or unsaturated fatty acids (FAs), we evaluated the areas of all the peaks identified on the basis of their molecular weights. A total of 24 species were investigated: 16:1 SM, 18:0 SM, 18:2 SM, 20:0 SM, 20:1 SM, 20:2 SM, 20:3 SM, 22:0 SM, 22:1 SM, 22:2 SM, 22:3 SM, 22:4 SM, 24:1 SM, 24:2 SM, 24:3 SM, 24:4 SM, 24:5 SM, 26:0 SM, 26:1 SM, 26:2 SM, 26:3 SM, 26:4 SM, 26:5 SM, 26:6 SM. 8 peaks were detected (Figure 4d panel A and B). Highly significant decrease for saturated intermediate-length acyl chains as 18:0 SM, 20:0 SM, and 22:0 SM was found in 1α,25(OH)_2_D_3_ treated U251 and LN18 cells in comparison with untreated cells (Figure 4d).

By comparing the changes in the total level of saturated and unsaturated SM species, we obtained that unsaturated/saturated SM ratio was 1.2 in in untreated sample and 1.49 in 1α,25(OH)_2_D_3_ treated. 

Since the data above suggested that 1α,25(OH)_2_D_3_ upregulated and activated nSMase1 in U251 cells, it became important to determine the consequences of this. Our laboratory previously reported that 1α,25(OH)_2_D_3_ treatment of embryonic hippocampal cells resulted in the cell differentiation via nSMase [16]. Therefore, differentiation of U251 and LN18 cells was considered. We performed immunofluorescence analysis to label glial fibrillary acidic protein (GFAP), as glial marker differentiation in U251 [34] and LN18 [35] cells. An increase in cell differentiation with respect to the control sample with the formation of numerous neurites in both U251 and LN18 cells was evident; the fluorescent intensity increased 2.13 and 3.25 times in U251 and LN18 cells, respectively (Figure 5a,b).

Finally, it has been shown that 1α,25(OH)_2_D_3_ VD3 modulated embryonic hippocampal cell differentiation thanks to the slowdown of cell growth [18]. Therefore, to evaluate the role of 1α,25(OH)_2_D_3_ in U251 cell growth we used an in vivo experiment by performing a xenotransplantation of U215 cell treated with 400 nM 1α,25(OH)_2_VD3. As can be seen, chorioallantoic membrane (CAM) had good visible blood vasculature useful for the success of transplantation (Figure 6a) and the engraftment of the transplant was present (Figure 6b). Ki-67 positive cells were evident in untreated U215 cells cultured in CAM (CTR, Figure 6c). If the cells were treated with 400 nM 1α,25(OH)_2_VD3, the labeling reduced 59% (Figure 6c,d), by indicating a reduction of proliferative index [26]. To confirm these data, the gene expression of CCND1, encoding for Cyclin D1 protein responsible for G1/S phase transition of the cell cycle, was measured. CCND1 expression was evaluated in U251 cells cultured in the presence of 400 nM 1α,25(OH)_2_D_3_ VD3 and transplanted in the CAM. As expected, 400 nM 1α,25(OH)_2_D_3_ VD3 was able to downregulate CCND1gene expression (Figure 6e), by indicating a reduction of cell growth [27].

Taken together, these results demonstrate that 1α,25(OH)_2_VD_3_ plays an important role in the reduction of U251 cell growth and in the induction of their differentiation via non genomic pathway with the involvement of SM metabolism. 

## 3. Discussion

The relevance of SM breakdown in cancer cell fate underscored by the role of aSMase in apoptosis modulation and of nSMase in cell differentiation modulation [18,26,28] became a determinant for us to investigate its behavior in GBM cells treated with 1α,25(OH)_2_VD3.

Although the action of 1α,25(OH)_2_VD3 has been implicated in different pathophysiological conditions [36,37,38] and its action via nSMase and aSMase has been described [17,18,19], its function and its involvement in aSMase and nSMase gene and protein expression in GBM cells have not been investigated yet. In this study, we demonstrate that WT p53 cells respond to a physiological dose of 1α,25(OH)_2_VD3 with upregulation of its receptor without the involvement of SM breakdown, by indicating the activation of classical via 1α,25(OH)_2_VD3/VDR. It is not possible at the moment to explain why VDR is not overexpressed when we use a high dose of 1α,25(OH)_2_VD3. We can speculate that high dose of vitamin induces mechanisms of regulation to limit a response that would otherwise be exaggerated for the cells. Importantly, we present a cell-specific role for 1α,25(OH)_2_VD3 in GBM cells. In fact, mutant p53 U251 and LN18 cells do not respond to 1α,25(OH)_2_VD3 with classical via VDR-mediated but via SMase1. This result is very important considering that the expression of mutant p53 is characteristic of the most GBs and it is implicated in cell invasion, migration, proliferation, evasion of apoptosis of GBM cells [39]. Therefore, mutant p53 was considered useful as a diagnostic marker and/or target for innovative therapeutic strategies [39]. 

Our data show that high dose of vitamin upregulates gene and protein expression of nSMase1, an enzyme that stimulates cell differentiation [22].

Previous studies have revealed significant roles played by 1α,25(OH)_2_VD3 in GBM. However, depending on cell type, 1α,25(OH)_2_VD3 could function or not. In fact, Tx3095, Tx3868, U87, U118, and U373 GBM cell lines were resistant against the antiproliferative activity of 1 α,25(OH)_2_VD_3_ [9], and U251, T98G [10,11], and U87-MG [12] GBM cell lines were susceptible. Therefore, despite different cellular responses of various GBM cell types to 1α,25(OH)_2_VD_3_ have been reported, the dependence on cell type for the genomic or non-genomic 1α,25(OH)_2_VD_3_ response has not been reported yet.

The major conclusion from this study relates to the mechanism of 1 α,25(OH)_2_VD3 action in p53 wild-type GL15 and mutant p53 nt U251 and LN18 cells. First, the results show that the response of U251 and U251 cells correlates with nSMase1 upregulation. The results of this study show that this induction is directly dependent on the 1 α,25(OH)_2_VD3 treatment. Thus, we propose that cells with p53 mutation select for the overexpression of nSMase1, which then imparts a growth delay and differentiation to those cells. Thus, though 1α,25(OH)_2_VD3 pathophysiology has a complex and not yet well-understood mechanism in GBM growth slowdown and differentiation, our model indicates that SM breakdown is involved in this process. It is possible to speculate on the possible involvement in this process of both ceramide, derived from SM via SMase1, and of the increase in unsaturated/saturated SM ratio. This is clearly illustrated and corroborated by the gain-of-function studies that showed how ceramide is involved in cell division and differentiation [40,41]. Notably, ceramide acts as tumor suppressor [42] and for this reason has attracted attention for cancer treatment [43]. It is not currently known whether ceramide can reduce resistance to chemotherapy drugs. However, it has been shown that ceramide-rubusoside nanomicelles can be a new therapeutic approach to target cancers carrying p53 missense mutations [44]. Additionally, Madigan et al. (2020) demonstrated an important cellular role for glucosylceramide synthase in oxaliplatin chemosensitivity [45]. 

Moreover, it is possible to hypothesize that the decrease in saturated and the increase in unsaturated SM species might be responsible for the increase in membrane fluidity of the cells and determine their enhanced dynamic properties that might facilitate the neurite plasticity [46]. Future studies could be useful to understand the changes in the physical state of cell membranes (i.e., cell and nuclear membranes) following SM and Cer changes induced by vitamin D3. Moreover, due to high molecular heterogeneity of the GBM in patients, nSMase1 might be a useful guide to establish GBM prognosis and precision treatment strategies in patients with mutant p53 which could benefit from the use of 1α,25(OH)_2_VD3 as a therapeutic adjuvant.

## 4. Materials and Methods

### 4.1. Chemicals

Cell culture medium (Dulbecco’s modified minimum essential medium, DMEM), fetal bovine serum (FBS), penicillin G, streptomycin, and glutamine, and sodium pyruvate were from GIBCO Invitrogen (Carlsbad, CA, USA). Anti-GFAP antibody was obtained from Dako, Agilent (Santa Clara, CA, USA), anti-VDR from Elabscience (Houston, TX, USA), anti-nSMase1 was from Santa Cruz Biotechnology, Inc. (Santa Cruz, CA, USA), anti-aSMase was from Abcam (Cambridge, UK), anti-β-tubulin and 1α,25(OH)_2_VD3, 3-[4,5-dimethyl-2-thiazolyl]-2,5-diphenyl-2-tetrazoliumbromide (MTT) were from Sigma Aldrich (St. Louis, MO, USA), horseradish peroxidase-conjugated goat anti-rabbit secondary antibodies and tetramethylrhodamineisothiocyanate (TRITC)-conjugated anti-rabbit IgG were from Santa Cruz. TaqMan SNP Genotyping Assay and Reverse Transcription kit were purchased from Applied Biosystems (Foster City, CA, USA). RNAqueous^®^-4PCR kit was from Ambion Inc. (Austin, TX, USA). SDS-PAGE molecular weight standards were purchased from Nzythech (Lisboa, Portugal). Chemiluminescence kitswerepurchasedfromAmersham (Rainham, Essex, UK).

### 4.2. Cell Culture

The GL15 cell line was established from human GBM and characterized by Bocchini et al. [47], LN18 and U251 GBM cell lines were purchased, respectively, from ATCC (ATCC^®^ CRL-2610, Manassas, VA, USA) and CLS Cell Lines Service GmbH (cat. n. 300385, Hamburg, Germany).

Cells were grown in DMEM supplemented with 10% heat-inactivated fetal bovine serum, 100 IU/mL penicillin/streptomicyn, and 200 mM of L-glutammine. Cells were maintained at 37 °C in a saturating humidity atmosphere containing 95% air and 5% CO_2_. 100 nM or 400 nM VD3 was added to the culture medium for 24 h (gene and protein expression) or 400 nM VD3 for 3 days (immunofluorescence). 

### 4.3. Cytotoxicity 

MTT assay was used to test cellular cytotoxicity, as previously reported [48]. Control and VD3 treated GL15 and U251 cells were seeded into 96-well plates at a density of 1 × 10^4^ cells/well with DMEM complete medium. After 24 h, culture medium was replaced with fresh complete medium containing 100 and 400 nM VD3, and the cells were incubated for 24 h. Then, MTT reagent was dissolved in PBS 1X and added to the culture at 0.5 mg/mL final concentration. After 3 h of incubation at 37 °C, the supernatant was carefully removed, and formazan salt crystals were dissolved in 200 µL DMSO that was added to each well. The absorbance (OD) values were measured spectrophotometrically at 540 nm using an automatic microplate reader (Eliza MAT 2000, DRG Instruments, GmbH, Marburg, Germany). Each experiment was performed two times in triplicate and the results were expressed as a percentage relative to the control cells. 

### 4.4. Reverse Transcription Quantitative PCR (RTqPCR)

Total RNA was extracted from GL15, U251, and LN18 cells, and CAM containing transplanted U251 cells by using RNAqueous-4PCR kit, as previously reported [25]. Before cDNA synthesis, the integrity of RNA was evaluated by electrophoresis in TAE 1.2% agarose gel. cDNA was synthesized using 1μg total RNA for all samples by High-Capacity cDNA Reverse Transcription kit under the following conditions: 50 °C for 2 min, 95 °C for 10 min, 95 °C for 15 s and 60 °C for 1 min, for a total of 40 cycles [28]. The following target genes were investigated: SMPD1 (Hs03679347_g1), SMPD4 (Hs04187047_g1), VDR (Hs00172113_m), and CCND1 (HS00765553); GAPDH (Hs99999905_m1), 18S rRNA (S18, Hs99999901_s1) and TUBB (Hs00742828_s1) were used as a housekeeping genes. mRNA relative expression levels were calculated as 2^−ΔΔCt^ comparing the results of the treated samples with those of the untreated ones [25].

### 4.5. Protein Concentration and Western Blotting

Protein concentration was measured as previously reported [25]. Proteins (40 μg) were submitted to 12% SDS (sodium dodecyl sulfate)-polyacrylamide gel electrophoresis at 200 V for 60 min [28]. Briefly, proteins were transferred onto 0.45 μm cellulose nitrate strips membrane (Sartorius StedimBiotech S.A.) in transfer buffer for 1 h at 100 V at 4 °C. Membranes were blocked with 5% (*w/v*) non-fat dry milk in PBS, pH7.5 for 1 h at room temperature. The blot was incubated overnight at 4 °C with specific antibodies (1:1000) and then treated with horseradish peroxidase-conjugated goat anti-rabbit secondary antibodies (1:5000). Super Signal West Pico Chemiluminescent Substrate (Thermo Fisher Scientific, Monza, Italy) was used to detect chemiluminescent (ECL) Horseradish peroxidase (HRP) substrate. The apparent molecular weight of proteins was calculated in relation to the migration rate of molecular size standards. The area density of the bands was evaluated by densitometry scanning and analyzing them with ImageJ.

### 4.6. Enzyme Activity Assay

nSMase activity was assayed as previously reported [25]. U251 cells were suspended in 0.1% NP-40 detergent in PBS, sonicated for 30 s on ice at 20 watt, kept on ice for 30 min and centrifuged at 16,000× *g* for 10 min. The supernatants were used for nSMase assay. The enzyme activity was assayed in 60 µg proteins/10 µL Tris-MgCl_2_, pH 7.4 using Amplex Red Sphingomyelinase assay kit (Invitrogen, Monza, Italy) according to the manufacturing instructions. The fluorescence was measured with FLUOstar Optima fluorimeter (BMG Labtech, Offemburg, Germany), by using the filter set of 360 nm excitation and 460 nm emission.

### 4.7. Ultrafast Liquid Chromatography–Tandem Mass Spectrometry

Lipid extraction and Ultrafast liquid chromatography–tandem mass spectrometry were performed according to Lazzarini et al., 2015 [49]. The 12:0 SM, 16:0 SM, 18:1 SM, 24:0 SM, 16:0 ceramide, 20:0 ceramide, and 24:0 ceramide standards were prepared as previously reported [47]. Standards were dissolved in chloroform/methanol (9:1 vol/vol) at 10 μg/mL final concentration. The stock solutions were stored at −20 °C. Working calibrators were prepared by diluting stock solutions with methanol to 500:0, 250:0, 100:0, and 50:0 ng/mL final concentrations. Twenty microliters of standards or lipids extracted from serum was injected after purification with specific nylon filters (0.2 μm). Analyses were carried outby using the Ultra Performance Liquid Chromatography system tandem mass spectrometer (Applied Biosystems, Monza, Italy). The lipid species were separated, identified, and analyzed as previously reported [49,50]. The samples were separated on a PhenomenexKinetex phenyl-hexyl 100 A column (50 × 4.60-mm diameter, 2.6-μm particle diameter) with a precolumn security guard Phenomenex ULTRA phenyl-hexyl 4.6. For SM, column temperature was set at 50 °C and flow rate at 0.9 mL/min. Solvent A was 1% formic acid; solvent B was 100% isopropanol containing 0.1% formic acid. The run was performed for 3 min in 50% solvent B and then in a gradient to reach 100% solvent B in 5 min. The system needed to be reconditioned for 5 min with 50% solvent B before the next injection. The lipid species were identified by using positive turbo-ion spray and modality multipole-reaction monitoring. 

### 4.8. Xenotrasnplantation in Chorioallantoic Membrane

Xenotransplantation was performed according to Uematsu et al. [51] with modifications. Fertilized chicken eggs were incubated at 37 °C under 52% humidity. On the third day, each egg was washed with 70% ethanol, after which a circular window was cut into the pointed pole of the shell, and 2–3 mL of albumen was removed to allow for the detachment of the developing chorioallantoic membrane (CAM). The window was sealed with a plastic film, and the eggs were put back into the incubator. On the eighth day, the window was enlarged to approximately 10 × 10 mm to reveal the underlying embryo and CAM vessels. A Dearling support was placed in the upper CAM, inside which U231 cells (400,000 cells) not treatred or treated for 24 h with 400 nM VD3were transplanted. Thereafter, the shell window was again sealed, and the eggs were returned to the incubator. After 12 days, cells were photographed in ovo with a stereomicroscope equipped with a camera, and an image analyser system (Leika). Then the grafts were fixed in ovo with 4% paraformaldehyde, the CAM was removed and dropped with essentially random orientation in paraffin. The paraffin blocks were sectioned into 4-mm-thick sections in a perpendicular plane to the CAM surface. All sections were mounted on silan-coated glass slides. Each slide contained a pair of sections at a distance equal to 140 mm. Six pairs of sections were sampled excluding the first and the last, and used for immunohistochemical analysis. 

### 4.9. Immunofluorescence

GFAP immunofluorescence analyses was performed as previously described [52]. Cells were incubated with anti-GFAP primary antibody diluted 1:100 in 3% (*w*/*v*) BSA in PBS for 1 h, washed three times in 0.1% (*v*/*v*) Tween-20 in PBS and twice in PBS, incubated with tetramethylrhodamineisothiocyanate (TRITC)-conjugated antirabbit IgG for 1 h, diluted 1:50 in 3% (*w*/*v*) BSA in PBS and washed as above. The diamidino-2-phenylindole (DAPI) nuclear counterstain was used. The samples were mounted in 80% (*w*/*v*) glycerol, containing 0.02% (*w*/*v*) NaN3 and p-phenylenediamine (1 mg/mL) in PBS to prevent fluorescence fading. The antibody incubations were done in a humid chamber at room temperature. Fluorescent analysis was performed on a DMRB Leika epi-fluorescent microscope equipped with a digital camera. The analysis of the tissue section size was performed by Scion Image software [52].

### 4.10. Immunohistochemistry

For immunohistochemical analysis Bond Dewax solution was used for the removal of paraffin from tissue sections before rehydration and immunostaining for Ki-67 (MIB-1 clone) on the Bond automated system (Leica Biosystems Newcastle Ltd., Upon Tyne, UK) as previously reported [52]. The observations were performed by using inverted microscopy EUROMEX FE 2935 (ED Amhem, The Netherland) equipped with a CMEX 5000 camera system (20× magnification). The analysis of the tissue section size was performed by ImageFocus software (this is a software of the CMEX 5000 with unic version) [53].

### 4.11. Statistical Analysis

Three independent experiments performed in duplicate were carried out for each analysis. Data are expressed as mean ± SD; Student’s t test was used for statistical analysis 1α,25(OH)_2_VD3-treated samples against the control sample (CTR).

## 5. Conclusions

In conclusion, our data present new insights into the involvement of SM breakdown in the 1α,25(OH)_2_VD3 treatment of U251 cells that might be useful to understand the use of vitamins in combination therapy for GBM treatment, as the ceramide produced could positively modulate the therapeutic response. 

## Figures and Tables

**Figure 1 cancers-12-03163-f001:**
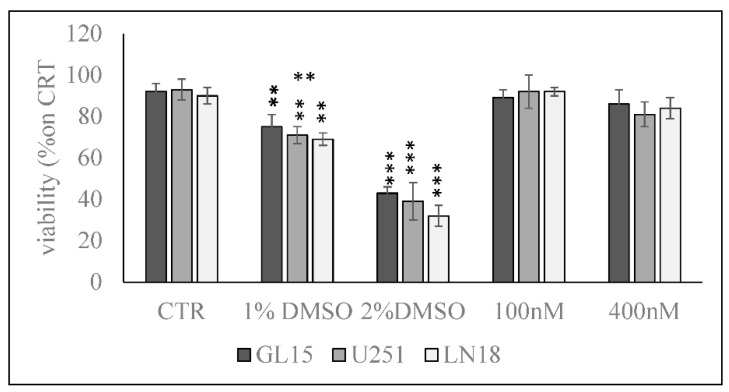
Effect of 1α,25(OH)_2_VD3 on GL15 U251, and LN18 cell viability. Cells were cultured with 100 and 400 nM 1α,25(OH)_2_VD3 for 24 h, and the viability was measured by MTT assay. Values are reported as the percent of viability compared to the control sample (CTR); 1% DMSO and 2% DMSO were used as positive controls. The data were expressed as mean ± SD of three independent experiments performed in duplicate. Significance, ∗∗ *p* < 0.01, ∗∗∗ *p* < 0.001 versus the CTR sample.

**Figure 2 cancers-12-03163-f002:**
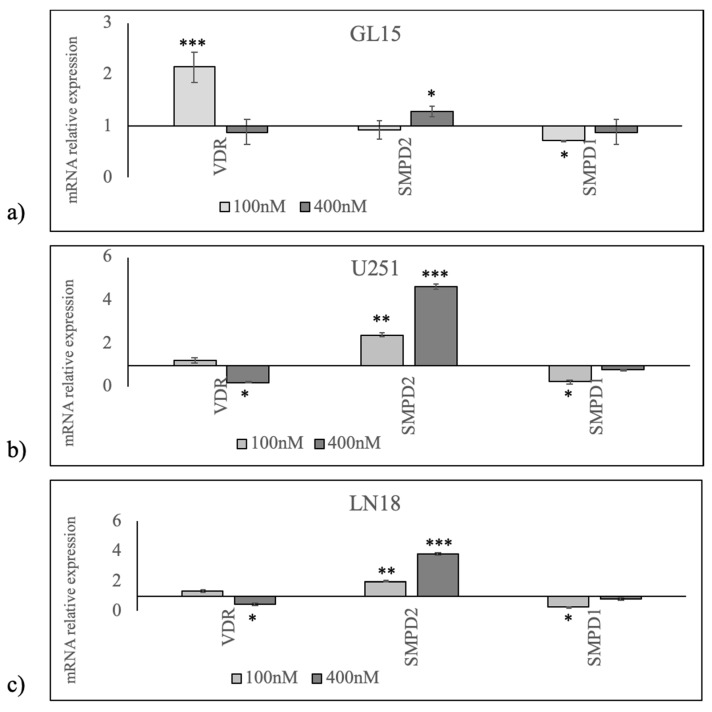
Effect of 100 and 400 nM 1α,25(OH)_2_VD_3_ on vitamin D receptor (VDR), *SMPD2* (encoding for neutral sphingomyelinase1), and *SMPD1* (encoding for acid sphingomyelinase) gene expression. (**a**) In GL15 cells, (**b**) U251 cells, and (**c**) LN18 cells, reverse transcription quantitative PCR (RTqPCR) analysis was performed in untreated (CTR) and 1α,25(OH)_2_VD_3_ treated cells. GAPDH was used as housekeeping gene. mRNA relative expression levels were calculated as 2^−ΔΔCt^ respect to the CTR sample. Data are expressed as the mean ± S.D. of 3 independent experiments performed in duplicate. ∗ *p* < 0.05, ∗∗ *p* < 0.01, ∗∗∗ *p* < 0.001.

**Figure 3 cancers-12-03163-f003:**
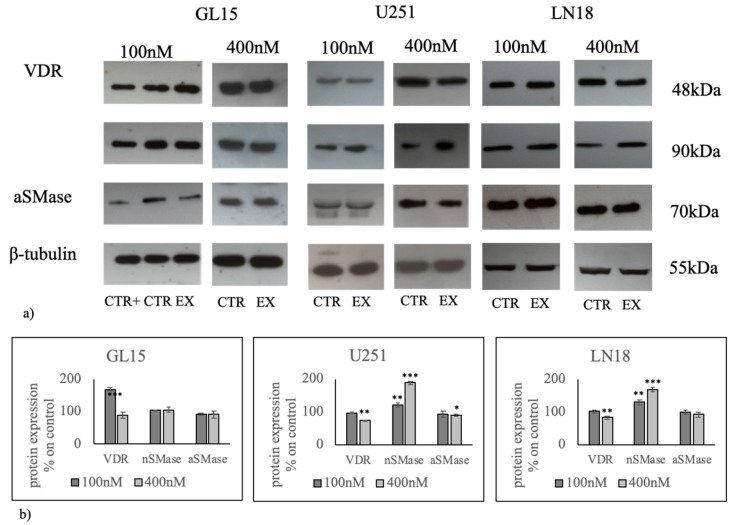
Effect of 100 nM and 400 nM 1α,25(OH)_2_D_3_ on vitamin D receptor (VDR), neutral sphingomyelinase (nSMase), and acid sphingomyelinase (aSMase) protein regulation in GL15 and U251 cells. (**a**) Western blotting image, β-tubulin was used as loading control; (**b**) densitometric analysis of immublotting bands. As a positive control (CTR+), embryonic hippocampal cells were used [33,34]. The values were first normalized with those of β-tubulin bands and then calculated as the percentage of CTR sample. Data are expressed as the mean ± S.D. of 3 independent experiments performed in duplicate. ∗ *p* < 0.05, ∗∗ *p* < 0.01, ∗∗∗ *p* < 0.001.

**Figure 4 cancers-12-03163-f004:**
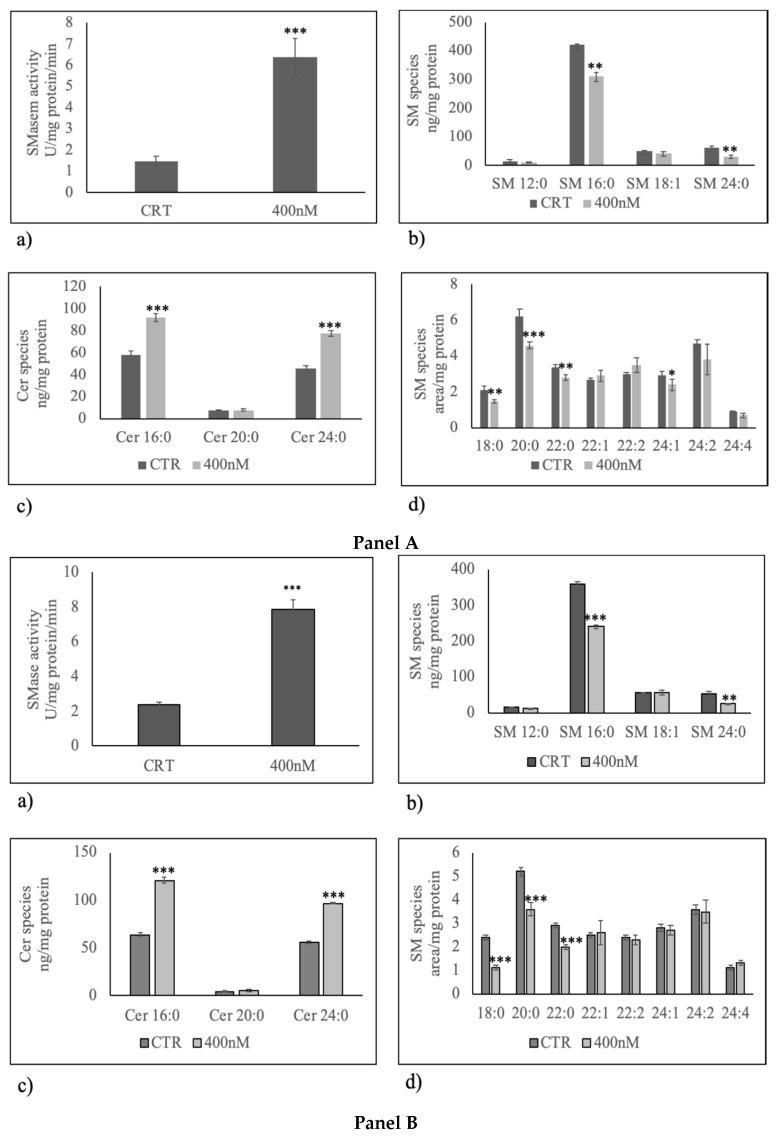
Effect of 400 nM 1α,25(OH)_2_D_3_ on sphingomyelin (SM) breakdown. Panel A, U251 cells; Panel B, LN18 cells. (**a**) Sphingomyelinase activity. The enzyme activity was evaluated by using the Amplex Red Sphingomyelinase assay kit, as reported in the “Materials and Methods” section. Data were expressed as mU/mg protein/min; (**b**) SM species studied by using 16:0 SM, 18:1 SM, and 24:0 SM external calibrators. Data are expressed as nmol/mg protein; (**c**) ceramide species studied by using Cer 16:0, Cer 20:0, Cer 24:0 external calibrators. Data are expressed as nmol/mg protein; (**d**) SM species evaluating the areas of all the peaks identified on the basis of their molecular weight. Data are expressed as area/mg protein. Data are expressed as the mean ± S.D. of 3 independent experiments performed in duplicate. ∗ *p* < 0.05, ∗∗ *p* < 0.01, ∗∗∗ *p* < 0.001.

**Figure 5 cancers-12-03163-f005:**
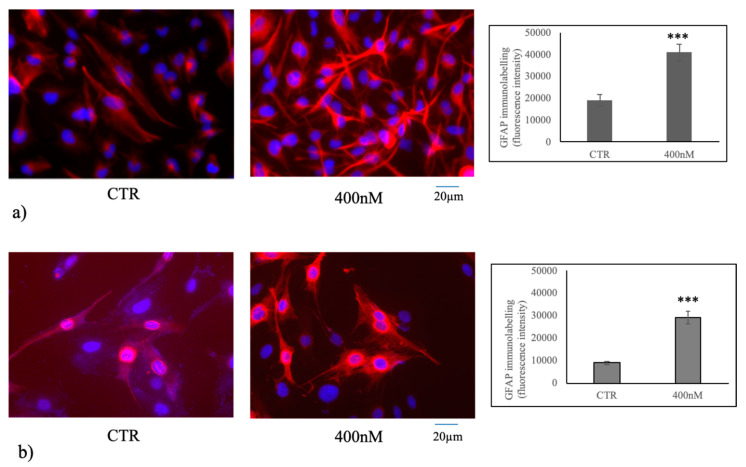
Immunofluorescence of U251 and LN18 cells without (CTR) or with 400 nM 1α,25(OH)_2_D_3_. The image represents the merged signal of glial fibrillary acidic protein (GFAP) immunolabelling, in red, counterstained with DAPI (in blue) in U251 cells (**a**) and LN18 cells (**b**). 20× magnification. The fluorescent intensity measurement is on the right. Data were expressed as the mean ± S.D. of 3 independent experiments performed in duplicate. The significance value of 400 nM 1α,25(OH)_2_D_3_ VD3 was calculated respect to the control sample. ∗∗∗ *p* < 0.001.

**Figure 6 cancers-12-03163-f006:**
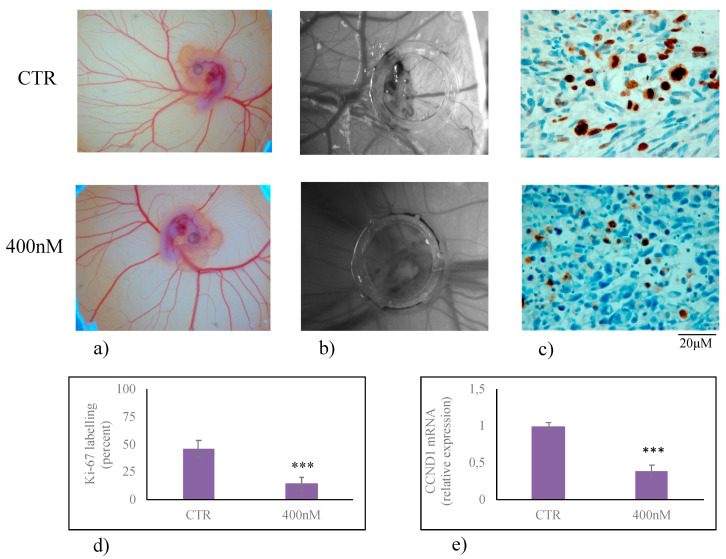
Xenotransplantation in chorioallantoic membrane (CAM) of U251 cells treated or not with 400 nM 1α,25(OH)_2_VD_3_. Experiments were performed as reported in the material and methods. (**a**) Image of embryo and CAM vessels on the eighth day from fertilization; (**b**) U231 cells photographed in ovo after 12 days from transplantation; (**c**) immunohistochemistry analysis of U231 cells in CAM fixed in ovo after 12 days from transplantation; (**d**) percentage of labelling; (**e**) CCND1gene expression performed by RTqPCR analysis. Data were expressed as the mean ± S.D. of 3 independent experiments performed in duplicate. ∗∗∗ *p* < 0.001.

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
