# Peer review of "Effect of 1α,25(OH)_2_ Vitamin D_3_ in Mutant P53 Glioblastoma Cells: Involvement of Neutral Sphingomyelinase1"

_cancers, 2020, doi:10.3390/cancers12113163_

Round 1

Reviewer 1 Report

I thank the authors for addressing all my comments and for the tremendous effort with providing an additional cell line. I do not have any further comments.

Reviewer 2 Report

The authors have replied satisfactorily to the reviewer's comments

Reviewer 3 Report

The authors address part of the issues raised in the previous version of the manuscript, in particular, that one referred to cytotoxicity. However, I am satisfied with the manuscript now. It shall proceed for publishing.

This manuscript is a resubmission of an earlier submission. The following is a list of the peer review reports and author responses from that submission.

Round 1

Reviewer 1 Report

The paper is well carried out, and the data presented are really interesting. However, there are some issues regarding the cell differentiation elicited by VD3 that the authors should address.

  1. Why the authors did note measure also the stem cell markers such as CD 113, CD44 or nestin, in order to have a ratio with GFAP?
  2. Immunofluorescence and immunohistochemistry studies, need to be confirmed by cytotoxicity, in order to affirm that the arrest of the proliferation occurred through differentiation processes
  3. Why the authors did not perform an evaluation of a lipid phase transition using VD3? 
  4. Why the authors choose only two GBM cell lines? Primary GBM cells would be a better and accurate model to study pathology.

Author Response

Reviewer 1

The paper is well carried out, and the data presented are really interesting. However, there are some issues regarding the cell differentiation elicited by VD3 that the authors should address.

Thank you so much for the truly constructive feedbak you have given us

  1. Why the authors did note measure also the stem cell markers such as CD 113, CD44 or nestin, in order to have a ratio with GFAP?

thanks for this observation! From the analysis of the bibliography we believed that CD 113, CD44 or nestin were mesenchymal and non-neuronal markers. This was our mistake.  In continuing our work, we will definitely consider these markers. Thanks again!

  1. Immunofluorescence and immunohistochemistry studies, need to be confirmed by cytotoxicity, in order to affirm that the arrest of the proliferation occurred through differentiation processes.

at the beginning of the work we had studied the cytotoxicity at 24 hours after treatment to see if VD3 had a toxic effect on the cells at the doses used in the study. We had not entered this result because the two doses of vitamin D used for the study did not induce cytotoxicity. But it is an important result and, therefore, we have entered it in the paper (Lines 109-113, Fig.1 and in “methods” lines 298,299, 315-325). Thanks again. Unfortunately we do not have data after three days of cell culture used for immunofluorescence. Now the cell culture laboratory is still closed due to Covid, further experiments are not possible. I'm sorry. In the paper, the effect of reduced proliferation was evident in vivo experiment by immunohistochemistry. Thus we had done the RT-PCR analysis of CCND1gene encoding for Cyclin D1 responsible for G1/S transition of the cell cycle. Thus, we have included this result (lines 216-221, Fig.6e and in “methods” line 334).  Thank you for this idea that really improved our results.

  1. Why the authors did not perform an evaluation of a lipid phase transition using VD3? 

because the objective of the study was to analyze the effect of VD3 on the metabolism of sphingomyelin and for this the sphingomyelinases were analyzed and a lipidomic study was carried out on the sphingomyelin and ceramide. The study was performed on whole cells. It is possible that the sphingomyelin and ceramide modification also induce a change in the physical state of the membranes. This is also an excellent idea. We could in a future study analyze the effect of VD3 on the physical state of the cell membrane and nuclear membrane, the methods of which we already have in the laboratory. Thanks again for the tip. It has been included in the “discussion” section (lines 284-286).

  1. Why the authors choose only two GBM cell lines? Primary GBM cells would be a better and accurate model to study pathology.

because we have been working with these two cell lines for a long time and therefore we knew them very well. However your observation is very fair, thank you very much, we will keep this observation in our future studies. As above reported, at the moment the laboratory of cell culture is closed and it is not possible to perform the experiments

Reviewer 2 Report

The manuscript in its present form is not acceptable, it has serious flaws in experimental design and contradicts the results at many places.

Author Response

The manuscript in its present form is not acceptable, it has serious flaws in experimental design and contradicts the results at many places.

We have made all the corrections requested by the other referees which have greatly improved the work

Reviewer 3 Report

The ms by Cataldi et al demonstrates a Sphingomyelin/Ceramide involvement in the effect of 1alpha,25(OH)2 Vit D3 as prodifferentiation and antiproliferative signaling molecule in glioblastoma cells.

The ms is well designed and the results support the conclusions drawn by the authors.

Minor comments:

In most figures, the y-axis is not enough explaining the significance of the data reported.

Fig. 2b: y-axis: use the expression "Protein expression (% on control)" instead of "Percentage"

Fig. 3a: y-axis: use "SMase activity (mU/mg protein/min)" 

Fig 3b and d: y-axes: use "SM species content (ng/mg protein)" and "SM species content (area/mg protein)"

Fig. 3c: Y-axis: use "Cer species content (ng/mg protein)"

Fig. 4b: y-axis: use "GFAP immunolabelling (fluorescent intensity)"

Fig 5b: y-axis: use "Immunoblotting (percent)"

Fig 5 legend: Misprint --> Xenotransplantation, please correct

Please check references from 42 to 48. There is no correspondence with text.

Author Response

1alpha,25(OH)2 Vit D3 as prodifferentiation and antiproliferative signaling molecule in glioblastoma cells.

The ms is well designed and the results support the conclusions drawn by the authors.

Minor comments:

In most figures, the y-axis is not enough explaining the significance of the data reported.

Fig. 2b: y-axis: use the expression "Protein expression (% on control)" instead of "Percentage"

It has been made, see Fig.3b in the present version

Fig. 3a: y-axis: use "SMase activity (mU/mg protein/min)" 

It has been made see Fig.4a in the present version

Fig 3b and d: y-axes: use "SM species content (ng/mg protein)" and "SM species content (area/mg protein)"

It has been made. see Fig.4b and d in the present version

Fig. 3c: Y-axis: use "Cer species content (ng/mg protein)"

It has been made. see Fig.4c in the present version

Fig. 4b: y-axis: use "GFAP immunolabelling (fluorescent intensity)"

It has been made. see Fig.5b in the present version

Fig 5b: y-axis: use "Immunoblotting (percent)"

It has been made. see Fig.6b in the present version

Fig 5 legend: Misprint --> Xenotransplantation, please correct

It has been corrected

Please check references from 42 to 48. There is no correspondence with text.

All references have been corrected. Thank you very much

Reviewer 4 Report

I thank the authors for this well-written paper investigating the effect of 1α,25(OH)2 vitamin D3 in p53 mutated U251 GBM cells. The authors used two cell lines, wild-type p53 GL15 and mutated p53 U251 and demonstrated that 1α,25(OH)2 vitamin D3 acts via vitamin D receptor in GL15 cells and via neutral sphingomyelinase1, with an enrichment of ceramide pool, in U251 cells. 

Minor: 

It is uncommon to report a summary of the results in the introduction. 

Major:

-The authors mentioned the role of cell line considering the impact of  1α,25(OH)2 vitamin D3 in the discussion. I would recommend to investigate another p53 mutated GBM cell line to increase the scientific impact of the manuscript, as the authors make the p53 mutation responsible for the overexpression of nSMAse.

-The authors should include and discuss this study to highlight the role of p53 pathway in GBM:

The p53 Pathway in Glioblastoma

Ying Zhang,1, Collin Dube,1, Myron Gibert, Jr.,1, Nichola Cruickshanks,1 Baomin Wang,1 Maeve Coughlan,1 Yanzhi Yang,1 Initha Setiady,1 Ciana Deveau,1 Karim Saoud,1 Cassandra Grello,1 Madison Oxford,1 Fang Yuan,1 and Roger Abounader1,2,3,*

- In regard of the heterogeneity of the GBM in patients, I would kindly ask the authors to discuss and highlight the translational aspects of their findings better.

Author Response

I thank the authors for this well-written paper investigating the effect of 1α,25(OH)2 vitamin D3 in p53 mutated U251 GBM cells. The authors used two cell lines, wild-type p53 GL15 and mutated p53 U251 and demonstrated that 1α,25(OH)2 vitamin D3 acts via vitamin D receptor in GL15 cells and via neutral sphingomyelinase1, with an enrichment of ceramide pool, in U251 cells. 

thank you very much for your appreciation

Minor: 

It is uncommon to report a summary of the results in the introduction. 

The results have been deleted from the introduction

Major:

-The authors mentioned the role of cell line considering the impact of  1α,25(OH)2 vitamin D3 in the discussion. I would recommend to investigate another p53 mutated GBM cell line to increase the scientific impact of the manuscript, as the authors make the p53 mutation responsible for the overexpression of nSMAse.

You are right, this is a very good idea but the laboratory of cell culture is closed due to Covid and it is not possible at the moment to perform this experiments. I hope you understand this. In continuing our work, we will definitely consider your suggestion. Thanks again!

-The authors should include and discuss this study to highlight the role of p53 pathway in GBM:

The p53 Pathway in Glioblastoma

Ying Zhang,1,† Collin Dube,1,† Myron Gibert, Jr.,1,† Nichola Cruickshanks,1 Baomin Wang,1 Maeve Coughlan,1 Yanzhi Yang,1 Initha Setiady,1 Ciana Deveau,1 Karim Saoud,1 Cassandra Grello,1 Madison Oxford,1 Fang Yuan,1 and Roger Abounader1,2,3,*

Thank you very much, the interesting paper has been included (lines 252-255, 522-523)

- In regard of the heterogeneity of the GBM in patients, I would kindly ask the authors to discuss and highlight the translational aspects of their findings better.

It has been discussed at the end of “Discussion” section (lines 284-289). Thank you very much for this suggestion

Round 2

Reviewer 2 Report

I appreciate the efforts made by author in this article however, the results presented in the article are insufficient and inconclusive.

Author Response

the work has been considerably improved by adding all the experiments required by refereee 4

Reviewer 4 Report

I thank the authors for the revision and I understand the limitation due to Corona, however I think that the quality of the manuscript and the conclusion could be improved, if another cell-line could be added. Therefore, I would appreciate, if enough time in regard to corona limitations for this (still open) revision point could be provided to the authors.

Author Response

Reviewer 4

I thank the authors for this well-written paper investigating the effect of 1α,25(OH)2 vitamin D3 in p53 mutated U251 GBM cells. The authors used two cell lines, wild-type p53 GL15 and mutated p53 U251 and demonstrated that 1α,25(OH)2 vitamin D3 acts via vitamin D receptor in GL15 cells and via neutral sphingomyelinase1, with an enrichment of ceramide pool, in U251 cells. 

Minor: 

It is uncommon to report a summary of the results in the introduction. 

The summary of results had already been deleted in the previous review in response to your observation

Major:

-The authors mentioned the role of cell line considering the impact of  1α,25(OH)2 vitamin D3 in the discussion. I would recommend to investigate another p53 mutated GBM cell line to increase the scientific impact of the manuscript, as the authors make the p53 mutation responsible for the overexpression of nSMAse.

Thank you very much for your suggestion, cell viability, gene and protein expression, SMase activity, lipidomic and immunofluorescence analysis were conducted on the LN18 cell line (see results, materials and methods, and discussion)

-The authors should include and discuss this study to highlight the role of p53 pathway in GBM:

The p53 Pathway in Glioblastoma

Ying Zhang,1,† Collin Dube,1,† Myron Gibert, Jr.,1,† Nichola Cruickshanks,1 Baomin Wang,1 Maeve Coughlan,1 Yanzhi Yang,1 Initha Setiady,1 Ciana Deveau,1 Karim Saoud,1 Cassandra Grello,1 Madison Oxford,1 Fang Yuan,1 and Roger Abounader1,2,3,*

The article was included in relation to the results of our study (p. 13, lines 269-274, reference 39)

- In regard of the heterogeneity of the GBM in patients, I would kindly ask the authors to discuss and highlight the translational aspects of their findings better.

It has been included at the end of the “discussion section” (p.14, lines 304-309)
